# Beyond a Video Frame Interpolator: A Space Decoupled Learning Approach to Continuous Image Transition

**Abstract.** Video frame interpolation (VFI) aims to improve the temporal resolution of a video sequence. Most of the existing deep learning based VFI methods adopt off-the-shelf optical flow algorithms to estimate the bidirectional flows and interpolate the missing frames accordingly. Though having achieved a great success, these methods require much human experience to tune the bidirectional flows and often generate unpleasant results when the estimated flows are not accurate. In this work, we rethink the VFI problem and formulate it as a continuous image transition (CIT) task, whose key issue is to transition an image from one space to another space continuously. More specifically, we learn to implicitly decouple the images into a translatable flow space and a non-translatable feature space. The former depicts the translatable states between the given images, while the later aims to reconstruct the intermediate features that cannot be directly translated. In this way, we can easily perform image interpolation in the flow space and intermediate image synthesis in the feature space, obtaining a CIT model. The proposed space decoupled learning (SDL) approach is simple to implement, while it provides an effective framework to a variety of CIT problems beyond VFI, such as style transfer and image morphing. Our extensive experiments on a variety of CIT tasks demonstrate the superiority of SDL to existing methods. Codes will be made publicly available.

**Keywords:** Video Frame Interpolation, Continuous Image Transition, Image Synthesis, Space Decoupled Learning

## 1 Introduction

Video frame interpolation (VFI) targets at synthesizing intermediate frames between the given consecutive frames of a video to overcome the temporal limitations of camera sensors. VFI can be used in a variety of practical applications, including slow movie generation [26], motion deblurring [53] and visual quality enhancement [68]. The conventional VFI approaches [2] usually calculate optical flows between the source and target images and gradually synthesize the intermediate images. With the great success of deep neural networks (DNNs) in computer vision tasks [16, 21, 51], recently researchers have been focusing on developing DNNs to address the challenging issues of VFI.

Most DNN based VFI algorithms can be categorized into flow-based [26, 4, 67, 40], kernel-based [41, 32, 53], and phase-based ones [38, 37]. With the advancement of optical flow methods [58, 5], flow-based VFI algorithms have gained increasing popularity and shown good quantitative results on benchmarks [4, 40]. However, these methods require much human experience to tune the bidirectional flows, *e.g.*, by using the forward [26, 4] and backward [39, 40] warping algorithms. In order to improve the synthesis performance, some VFI methods have been developed by resorting to the depth information [4], the acceleration information [67] and the softmax splatting [40]. These methods, however, adopt the off-the-shelf optical flow algorithms, and hence they often generate unpleasant results when the estimated flows are not accurate.

To address the above issues, we rethink the VFI problem and aim to find a solution that is free of flows. Different from previous approaches, we formulate VFI as a continuous image transition (CIT) problem. It is anticipated that we could construct a smooth transition process from the source image to the target image so that the VFI can be easily done. Actually, there are many CIT tasks in computer vision applications, such as image-to-image translation [24, 69], image morphing [34, 45] and style transfer [19, 23]. Different DNN models have been developed for different CIT tasks. Based on the advancement of deep generative adversarial network (GAN) techniques [7, 28, 29], deep image morphing methods have been proposed to generate images with smooth semantic changes by walking in a latent space [48, 25]. Similarly, various image-to-image translation methods have been developed by exploring intermediate domains [20, 66, 14], interpolating attribute [36] or feature [60] or kernel [63] vectors, using physically inspired models for guidance [47], and navigating latent spaces with discovered paths [9, 25]. Though significant progresses have been achieved for CIT, existing methods usually rely on much human knowledge of the specific domain, and employ rather different models for different applications.

In this work, we propose to learn a translatable flow space to control the continuous and smooth translation between two images, while synthesize the image features which cannot be translated. Specifically, we present a novel space decoupled learning (SDL) approach for VFI. Our SDL implicitly decouples the image spaces into a translatable flow space and a non-translatable feature space. With the decoupled image spaces, we can easily perform smooth image translation in the flow space, and synthesize intermediate image features in the non-translatable feature space. Interestingly, the proposed SDL approach can not only provide a flexible solution for VFI, but also provide a general and effective solution to other CIT tasks.

To the best of our knowledge, the proposed SDL is the first flow-free algorithm which is however able to synthesize consecutive interpolations, achieving leading performance in VFI. SDL is easy-to-implement, and it can be readily integrated into off-the-shelf DNNs for different CIT tasks beyond VFI, serving as a general-purpose solution to the CIT problem. We conduct extensive experiments on various CIT tasks, including, VFI, image-to-image translation and image morphing, to demonstrate its effectiveness. Though using the same framework, SDL

shows highly competitive performance with those state-of-the-art methods that are specifically designed for different CIT problems.

## 2    Related Work

### 2.1    Video Frame Interpolation (VFI)

With the advancement of DNNs, recently significant progresses have been made on VFI. Long *et al.* [35] first attempted to generate the intermediate frames by taking a pair of frames as input to DNNs. This method yields blurry results since the motion information of videos is not well exploited. The latter works are mostly focused on how to effectively model motion and handle occlusions. Meyer *et al.* [38, 37] proposed phase-based models which represent motion as per-pixel phase shift. Niklaus *et al.* [41, 42] came up with the kernel-based approaches that estimate an adaptive convolutional kernel for each pixel. Lee *et al.* [32] introduced a novel warping module named Adaptive Collaboration of Flows (AdaCoF). An end-to-end trainable network with channel attention was proposed by Choi *et al.* [12], where frame interpolation is achieved without explicit estimation of motion. The kernel-based methods have achieved impressive results. However, they are not able to generate missing frames with arbitrary interpolation factors and usually fail to handle large motions due to the limitation of kernel size.

Unlike phase-based or kernel-based methods, flow-based models explicitly exploit motion information of videos [26, 4, 67, 40]. With the advancement of optical flow methods [58, 5], flow-based VFI algorithms have become popular due to their good performance. Niklaus and Liu [39] adopted forward warping to synthesize intermediate frames. This algorithm suffers from holes and overlapped pixels, and it was later improved by the softmax splatting method [40], which can seamlessly map multiple source pixels to the same target location. Since forward warping is not very intuitive to use, most flow-based works adopt backward warping. Jiang *et al.* [26] jointly trained two U-Nets [52], which respectively estimate the optical flows and perform bilateral motion approximation to generate intermediate results. Reda *et al.* [50] and Choi *et al.* [11] further improved this work by introducing cycle consistency loss and meta-learning, respectively. Bao *et al.* [4] explicitly detected the occlusion by exploring the depth information, but the VFI performance is sensitive to depth estimation accuracy. To exploit the acceleration information, Xu *et al.* [67] proposed a quadratic VFI method. Recently, Park *et al.* [44] proposed a bilateral motion network to estimate intermediate motions directly.

### 2.2    Continuous Image Transition (CIT)

In many image transition tasks, the key problem can be formulated as how to transform an image from one state to another state. DNN based approaches have achieved impressive results in many image transition tasks, such as image-to-image translation [24, 69, 62], style transfer [19, 27], image morphing [9] and VFI

[32, 42]. However, these methods are difficult to achieve continuous and smooth transition between images. A continuous image transition (CIT) approach is desired to generate the intermediate results for a smooth transition process.

Many researches on image-to-image translation and image morphing resort to finding a latent feature space and blending image features therein [60, 36, 47]. However, these methods need to explicitly define the feature space based on human knowledge of the domain. Furthermore, encoding an image to a latent code often results in the loss of image details. Alternatively, methods on image morphing and VFI first establish correspondences between the input images, for example, by using a warping function or bidirectional optical flows, to perform shape deformation of image objects, and then gradually blend images for smooth appearance transition [65, 33, 4, 40]. Unfortunately, it is not easy to accurately specify the correspondences, leading to superimposed appearance of the intermediate results. In addition to generating a continuous transition between two input images (source and target), there are also methods to synthesize intermediate results between two different outputs [23, 22].

**Image-to-image Translation:** Isola *et al.* [24] showed that the conditional adversarial networks (cGAN) can be a good solution to image-to-image (I2I) translation problems. Many following works, such as unsupervised learning [69], disentangled learning [31], few-shot learning [34], high resolution image synthesis [62], multi-domain translation [13], multi-modal translation [70], have been proposed to extend cGAN to different scenarios. Continuous I2I has also attracted much attention. A common practice to this problem is to find intermediate domains by weighting discriminator [20] or adjusting losses [66]. Some methods have been proposed to enable controllable I2I by interpolating attribute [36] or feature [60] or kernel [63] vectors. Pizzati *et al.* [47] proposed a model-guided framework that allows non-linear interpolations.

**Image Morphing:** Conventional image morphing methods mostly focus on reducing user-intervention in establishing correspondences between the two images [65]. Smythe [55] used pairs of mesh nodes for correspondences. Beier and Neely [6] developed field morphing utilizing simpler line segments other than meshes. Liao *et al.* [33] performed optimization of warping fields in a specific domain. Recently, methods [45, 1, 25] have been proposed to achieve efficient image morphing by manipulating the latent space of GANs [7, 29]. However, these methods often result in the loss of image details and require time-consuming iterative optimization during inference. Mao *et al.* [36] and Pizzati *et al.* [47] decoupled content and style spaces using disentangled representations. They achieved continuous style interpolations by blending the style vectors. However, these methods preserve the content of source image and they are not suitable to image morphing. Park *et al.* [45] overcame this limitation by performing interpolation in both the content and style spaces.

As can be seen from the above discussions, existing works basically design rather different models for different CIT tasks. In this work, we aim to develop a state decoupled learning approach to perform different CIT tasks, including VFI, image-to-image translation and image morphing, by using the same framework.

# 3 Proposed Method

## 3.1 Problem Formulation

Given a source image $I_0$ and a target image $I_1$, the goal of VFI is to synthesize an intermediate result $I_t$ from them:

$$I_t = \mathcal{G}(I_0, I_1, t), \qquad (1)$$

where $t \in (0, 1)$ is a control parameter and $\mathcal{G}$ is a transition mapping function.

To better preserve image details, researchers [4, 67, 40] have resorted to using bidirectional optical flows [58, 59] of $I_0$ and $I_1$, denoted by $F_{0 \rightarrow 1}$ and $F_{1 \rightarrow 0}$, to establish the motion correspondence between two consecutive frames. With the help of optical flows, $I_t$ can be obtained as follows:

$$I_t = \mathcal{G}(I_0, I_1, \mathcal{B}(F_{0 \rightarrow 1}, F_{1 \rightarrow 0}, t)), \qquad (2)$$

where $\mathcal{B}$ is a blending function. Forward [39, 40] and backward [4, 67] warping algorithms have been proposed to perform the blending $\mathcal{B}$ in Eq. (2).

The above idea for VFI coincides with some image morphing works [65, 33, 17], where the warping function, instead of optical flow, is used to mark the object shape changes in the images. However, it is not easy to specify accurately the correspondences using warping, resulting in superimposed morphing appearance. This inspires us to model VFI as a CIT problem and seek for a more effective and common solution.

One popular solution to CIT is to embed the images into a latent space, and then blend the image feature codes therein:

$$I_t = \mathcal{G}(\mathcal{B}(L_0, L_1, t)), \qquad (3)$$

where $L_0, L_1$ represent respectively the latent codes of $I_0, I_1$ in the latent space. For example, StyleGAN [28] performs *style mixing* by blending the latent codes at various scales. To gain flexible user control, disentangled learning methods [36, 34, 47] were later proposed to decompose the latent space into the content and style representations. The smooth style mixing can be achieved by interpolating the style vectors as follows:

$$I_t = \mathcal{G}(L_0^c, \mathcal{B}(L_0^s, L_1^s, t)), \qquad (4)$$

where $L_0^s, L_1^s$ are the style representation vectors of $L_0, L_1$, respectively, and $L_0^c$ is the content vector of $L_0$. In this case, $I_1$ serves as the "style" input and the content of $I_0$ is preserved. However, the above formulation is hard to use in tasks such as image morphing.

Though impressive advancements have been made, the above CIT methods require much human knowledge to explicitly define the feature space, while embedding an image into a latent code needs time-consuming iterative optimization and sacrifices image details.

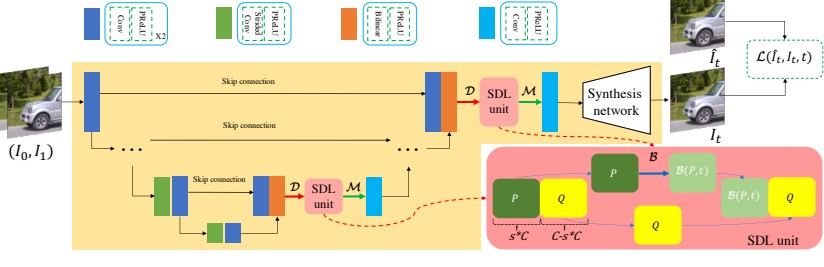

**Fig. 1.** The architecture of our space decoupled learning (SDL) method.

## 3.2   Space Decoupled Learning

As discussed in Section 3.1, previous works employ rather different models for different CIT applications. One interesting question is: can we find a common yet more effective framework to different CIT tasks? We make an in-depth investigation of this issue and present such a framework in this section.

The latent space aims to depict the essential image features and patterns of original data. It is expected that in the latent space, the correspondences of input images $I_0$ and $I_1$ can be well built. In other words, the latent codes $L_0, L_1$ in Eq. (3) play the role of optical flows $F_{0\to1}, F_{1\to0}$ in Eq. (2). Both of Eq. (3) and Eq. (2) blend the correspondence of two images to obtain the desired output. The difference lies in that the latent code representation of an image in Eq. (3) may lose certain image details, while in Eq. (2) the original inputs $I_0, I_1$ are involved into the reconstruction, partially addressing this problem.

From the above discussion, we can conclude that the key to CIT tasks is how to smoothly blend the image features whose correspondences can be well built, while reconstruct the image features whose correspondences are hard to obtain. We thus propose to decouple the image space into two sub-spaces accordingly: a *translatable flow space*, denoted by $P$, where the features can be smoothly and easily blended with $t$, and a *non-translatable feature space*, denoted by $Q$, where the features cannot be blended but should be synthesized. With $P$ and $Q$, we propose a unified formulation of CIT problems as follows:

$$I_t = \mathcal{G}(Q_{0\to1}, \mathcal{B}(P_{0\to1}, t)). \tag{5}$$

The subscript "$0 \to 1$" means the transition is from $I_0$ to $I_1$. With Eq. (5), we continuously transition those translatable image components in $P$, and reconstruct the intermediate features that cannot be directly transitioned in $Q$.

Now the question turns to how to define the spaces of $P$ and $Q$. Unlike many previous CIT methods [36, 47] which explicitly define the feature spaces using much human knowledge, we propose to learn $P$ and $Q$ implicitly from training data. We learn a decoupling operator, denoted by $\mathcal{D}$, to decompose the image space of $I_0$ and $I_1$ to the translatable flow space $P$ and the non-translatable feature space $Q$:

$$(P_{0\to1}, Q_{0\to1}) \leftarrow \mathcal{D}(I_0, I_1). \tag{6}$$

Specifically, we use several convolutional layers to implement the space decoupling operator $\mathcal{D}$. To gain performance, $\mathcal{D}$ is learned on multiple scales. The proposed method, namely space decoupled learning (SDL), requires no human knowledge of the domain, and it can serve as an effective and unified solution to different CIT tasks.

The architecture of SDL is a U-shaped DNN, as illustrated in Fig. 1. Unlike standard U-Net [52], a novel *SDL unit* is introduced in the decoder part of our network. The detailed structure of the SDL unit is depicted in the right-bottom corner of Fig. 1. The inputs of the SDL unit are the feature maps decomposed in previous convolution layers. Let $C$ be the number of input feature maps and $s \in (0,1)$ be the ratio of translatable flow features to the total features. $s$ is a hyper-parameter controlled by users (we will discuss how to set it in Section 4). We then split the channel number of input feature maps in $P$ and $Q$ as $s*C$ and $C - s*C$, and perform the blending $\mathcal{B}$ on $P$ while keeping $Q$ unchanged. There are multiple ways to perform the blending. For example, $\mathcal{B}$ can be achieved by scaling the features with factor $t$: $\mathcal{B}(P_{0 \to 1}, t) = t * P_{0 \to 1}$, which results in linear interpolation in $P$ and is used in our experiments. Afterwards, the blended $P$ and $Q$ are concatenated as the output of the SDL unit. A merging operator $\mathcal{M}$ (also learned as several convolutional layers like $\mathcal{D}$) is followed to rebind the decoupled spaces on multiple scales.

A synthesis network is also adopted to improve the final transition results. We employ a GridNet architecture [18] for it with three rows and six columns. Following the work of Niklaus *et al.* [40], some modifications are utilized to address the checkerboard artifacts. The detailed architecture of the synthesis network can be found in the **supplementary materials**. In addition, it is worth mentioning that $t$ works with the loss function during training if necessary. Details can be found in the section of experiments.

### 3.3   Training Strategy

To train SDL model for VFI, we adopt two loss functions: the Charbonnier loss [8] $\mathcal{L}_C$ and the perceptual loss [27] $\mathcal{L}_P$. The final loss $\mathcal{L}$ is as follows:

$$\mathcal{L} = \alpha \mathcal{L}_C + \beta \mathcal{L}_P, \tag{7}$$

where $\alpha$ and $\beta$ are balancing parameters. The content loss $\mathcal{L}_C$ enforces the fine features and preserves the original color information. The perceptual loss $\mathcal{L}_P$ can be better balanced to recover more high-quality details. We use the *conv5_4* feature maps before activation in the pre-trained VGG19 network [54] as the perceptual loss. In our experiments, we empirically set $\alpha = 1$ and $\beta = 0.1$.

For other CIT applications including image-to-image translation and image morphing, GAN plays a key role to generate high-quality results in order to alleviate superimposed appearances. In our implementation, we use PatchGAN developed by Isola *et al.* [24] for adversarial training. The final loss is the sum of the $\mathcal{L}_1$ loss and PatchGAN loss with equal weights.

**Table 1.** Quantitative comparison (PSNR, SSIM, runtime) of different methods on the Middleburry, UCF101, Vimeo90K and Adobe240fps datasets. The runtime is reported as the average time to process a pair of $640 \times 480$ images. The numbers in **bold** represent the best performance. The upper part of the table presents the results of kernel-based methods, and the lower part presents the methods that can perform smooth frame interpolations. "-" means that the result is not available.

| Method | Training Dataset | Runtime (ms) | Middleburry PSNR↑ | Middleburry SSIM↑ | UCF101 PSNR↑ | UCF101 SSIM↑ | Vimeo90K PSNR↑ | Vimeo90K SSIM↑ | Adobe240fps PSNR↑ | Adobe240fps SSIM↑ |
|---|---|---|---|---|---|---|---|---|---|---|
| SepConv [42] | proprietary | 57 | 35.73 | 0.959 | 34.70 | 0.947 | 33.79 | 0.955 | - | - |
| CAIN [12] | proprietary | 56 | 35.07 | 0.950 | 34.97 | 0.950 | 34.64 | 0.958 | - | - |
| AdaCof [32] | Vimeo90K | 77 | 35.71 | 0.958 | 35.16 | 0.950 | 34.35 | 0.956 | - | - |
| CDFI [15] | Vimeo90K | 248 | 37.14 | 0.966 | 35.21 | 0.950 | 35.17 | 0.964 | - | - |
| SuperSloMo [26] | Adobe240fps+Youtube240fps | 67 | 33.64 | 0.932 | 33.14 | 0.938 | 32.68 | 0.938 | 30.76 | 0.902 |
| DAIN [4] | Vimeo90K | 831 | 36.70 | 0.964 | 35.00 | 0.949 | 34.70 | 0.963 | 29.22 | 0.877 |
| BMBC [44] | Vimeo90K | 3008 | 36.78 | 0.965 | 35.15 | 0.950 | 35.01 | **0.965** | 29.56 | 0.881 |
| EDSC [10] | Vimeo90K-Septuplet | 60 | 36.81 | **0.967** | 35.06 | 0.946 | 34.57 | 0.956 | 30.28 | 0.900 |
| SDL | Vimeo90K+Adobe240fps | **42** | **37.38** | **0.967** | **35.33** | 0.951 | 35.47 | **0.965** | **31.38** | **0.914** |

**Fig. 2.** Visual comparison of competing methods on the Vimeo90K test set. (a) Sep-Conv [42]; (b) SuperSloMo [26]; (c) CAIN [12]; (d) EDSC [10]; (e) DAIN [4]; (f) BMBC [44]; (g) SDL; (h) Ground truth.

## 4    Experiments and Applications

In this section, we first conduct extensive experiments on VFI to validate the effectiveness of our SDL method, and then apply SDL to other CIT tasks beyond VFI, such as face aging, face toonification and image morphing, to validate the generality of SDL.

### 4.1    Datasets and Training Settings for VFI

There are several datasets publicly available for training and evaluating VFI models, including Middlebury [3], UCF101 [56], Vimeo90K [68] and Adobe240-fps [57]. The Middlebury dataset contains two subsets, *i.e.*, *Other* and *Evaluation*. The former provides ground-truth middle frames, while the later hides the ground-truth, and the users are asked to upload their results to the benchmark website for evaluation. The UCF101 dataset [56] contains 379 triplets of human action videos, which can be used for testing VFI algorithms. The frame resolution of the above two datasets is $256 \times 256$.

We combine the training subsets in Adobe240-fps and Vimeo90K to train our SDL model. The Vimeo90K dataset [68] has $51,312$ ($3,782$) triplets for training (testing), where each triplet contains 3 consecutive video frames of resolution $256 \times 448$. This implicitly sets the value of $t$ to 0.5, and hence it is insufficient to train our SDL model for finer time intervals. We further resort to the Adobe240-fps dataset [57], which is composed of high frame-rate videos, for model training. We first extract the frames of all video clips, and then group the extracted

frames with 12 frames per group. There is no overlap between any two groups. During training, we randomly select 3 frames $I_a, I_b, I_c$ from a group as a triplet, where $\{a, b, c\} \in \{0, 1, ..., 11\}$ and $a < b < c$. The corresponding value of $t$ can be calculated as $(b - a)/(c - a)$. We also randomly reverse the direction of the sequence for data augmentation ($t$ is accordingly changed to $1 - t$). Each video frame is resized to have a shorter spatial dimension of 360 and a random crop of $256 \times 256$. Horizontal flip is performed for data augmentation. Following SuperSloMo [26], we use 112 video clips for training and the rest 6 for validation.

During model updating, we adopt the Adam [30] optimizer with a batch size of 48. The initial learning rate is set as $2 \times 10^{-4}$, and it decays by a factor of 0.8 for every 100K iterations. The model is updated for 600K iterations.

### 4.2   Comparisons with State-of-the-arts

We evaluate the performance of the proposed SDL method in comparison with two categories of state-of-the-art VFI algorithms, whose source codes or pre-trained models are publicly available. The first category of methods allow frame interpolation at arbitrary time, including SuperSloMo [26], DAIN [4], BMBC [44] and EDSC [10]. The second category is kernel-based algorithms, including SepConv [42], CAIN [12], AdaCof [32] and CDFI [15], which can only perform frame interpolation iteratively at the power of 2. The PSNR and SSIM [64] indices are used for quantitative comparisons.

Table 1 provides the PSNR/SSIM and runtime results on the Middlebury *Other* [3], UCF101 [56], Vimeo90K [68] and Adobe240-fps [57] testing sets. In all experiments, the first and last frames of each group are taken as inputs. On the first three datsets, we set $t = 0.5$ to interpolate the middle frame. While on the high frame rate Adobe240-fps dataset, we vary $t \in \{\frac{1}{11}, \frac{2}{11}, ..., \frac{10}{11}\}$ to produce the intermediate 10 frames, which is beyond the capability of kernel-based methods [42, 12, 32, 15]. All the methods are tested on a NVIDIA V100 GPU, and we calculate the average processing time for 10 runs. From Table 1, one can see that the proposed SDL approach achieves best PSNR/SSIM indices on all the datasets, while it has the fastest running speed. The kernel-based method CDFI [15] also achieves very good PSNR/SSIM results. However, it often fails to handle large motions due to the limitation of kernel size. The flow-based methods such as DAIN [4] address this issue by referring to bidirectional flows, while inevitably suffer from inaccurate estimations. The proposed SDL implicitly decouples the images into a translatable flow space and a non-translatable feature space, avoiding the side effect of inaccurate flows.

Fig. 2 presents some visual comparisons of the VFI results of competing methods. It can be seen that our SDL method preserves better the image fine details and edge structures especially in scenarios with complex motions, where inaccurate flow estimations are commonly observed. SDL manages to address this difficulty by implicitly decoupling the images into a translatable flow space and a non-translatable feature space, and hence resulting in better visual quality with fewer interpolation artifacts. More visual comparison results can be found in the **supplementary material**. In the task of VFI, optical flow is widely used

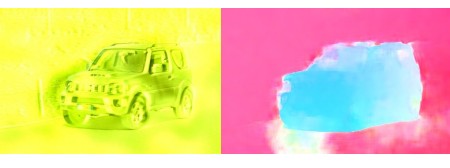 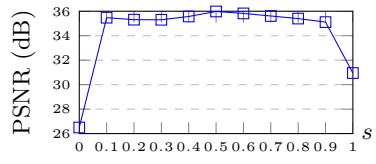

**Fig. 3.** Visualization of the translatable flow space and the optical flow in VFI. **Left:** the translatable flow space; **Right:** the optical flow.

**Fig. 4.** PSNR vs. $s$ on the Adobe240-fps testing set. When $s = 0.5$, the PSNR reaches the peak, while the performance is very stable by varying $s$ from 0.1 to 0.9.

**Table 2.** Quantitative comparison (PSNR, SSIM) between SDL and its variants on the Middlebury, UCF101, Vimeo90K and Adobe240fps datasets. The numbers in **bold** represent the best results.

| Method | Training Dataset | Middlebury PSNR↑ | SSIM↑ | UCF101 PSNR↑ | SSIM↑ | Vimeo90K PSNR↑ | SSIM↑ | Adobe240fps PSNR↑ | SSIM↑ |
|---|---|---|---|---|---|---|---|---|---|
| SDL-vimeo90k | Vimeo90K | **37.49** | **0.967** | 35.27 | **0.951** | **35.56** | **0.965** | 26.52 | 0.811 |
| SDL-w/o-sdl | Vimeo90K+Adobe240fps | 36.96 | 0.964 | 35.24 | 0.950 | 35.38 | 0.964 | 26.51 | 0.817 |
| SDL-w/o-syn | Vimeo90K+Adobe240fps | 37.19 | 0.965 | 35.27 | **0.951** | 35.37 | 0.964 | 31.21 | 0.911 |
| SDL | Vimeo90K+Adobe240fps | 37.38 | **0.967** | **35.33** | **0.951** | 35.47 | **0.965** | **31.38** | **0.914** |

to explicitly align the adjacent frames. However, this may lead to visual artifacts on pixels where the flow estimation is not accurate. In our SDL, we decouple the image space into a translatable flow space and a non-translatable feature space, and only perform interpolation in the former one, avoiding the possible VFI artifacts caused by inaccurate flow estimation. In Fig. 3, we visualize the the translatable flow space and compare it with the optical flow obtained by SpyNet [49]. As can be seen, the translatable flow space matches the optical flow on the whole, while it focuses more on the fine details and edge structures that are import to synthesize high-quality results.

## 4.3   Ablation Experiments

In this section, we conduct experiments to investigate the ratio of translatable flow features, and compare SDL with several of its variants.

**Translatable Flow Features.** In order to find out the effect of $s$ (*i.e.*, the ratio of translatable flow features to total features) of SDL, we set $s \in \{0, 0.1, ..., 1\}$) and perform experiments on the Adobe240-fps testing set. The curve of PSNR versus $s$ is plotted in Fig. 4. We can see that the performance decreases significantly if all feature maps are assigned to non-translatable feature space (*i.e.*, $s = 0$) or translatable flow space (*i.e.*, $s = 1$). When $s = 0.5$, the PSNR reaches the peak, while the performance is very stable by varying $s$ from 0.1 to 0.9. This is because SDL can learn to adjust its use of translatable and non-translatable features during training.

**The variants of SDL.** We compare SDL with several of its variants to validate the design and training of SDL. The first variant is denoted as SDL-vimeo90k, *i.e.*, the model is trained using only the Vimeo90K dataset. The second variant is denoted as SDL-w/o-sdl, *i.e.*, SDL without space decoupling learning

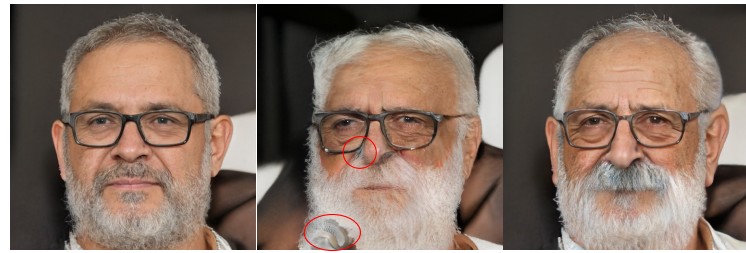

**Fig. 5.** Comparison of SDL with StyleGAN2 backpropagation on face aging. From left to right: input image, StyleGAN2 backpropagation [61] and SDL. Note that artifacts can be generated by StyleGAN2 backpropagation, while SDL can synthesize the image more robustly.

by setting $s = 0$. The third variant is denoted as SDL-w/o-syn, *i.e.*, the synthesis network is replaced with several convolution layers.

We evaluate SDL and its three variants on the Middlebury *Other* [3], UCF101 [56], Vimeo90K [68] and Adobe240-fps [57] testing sets, and the PSNR and SSIM results are listed in Table 2. One can see that SDL-vimeo90k achieves the best SSIM indices on all the triplet datasets, and the best PSNR indices on Middlebury *Other* and Vimeo90K by using a smaller training dataset than SDL, which uses both Vimeo90K and Adobe240-fps in training. This is because these is a domain gap between Adobe240-fps and Vimeo90k, and hence the SDL-vimeo90k can overfit the three triplet dataset. Furthermore, SDL-vimeo90k performs poorly on the Adobe240-fps dataset. This implies that training SDL using merely triplets fails to synthesize continuous frames.

Without decoupling the space, SDL-w/o-sdl performs much worse than the full SDL model, especially on the Adobe240-fps testing set. This validates that the space decoupling learning strategy boosts the VFI performance and plays a key role in continuous image transition. Without the GridNet [18], which is widely used as the synthesis network to improve VFI performance [39, 40], SDL-w/o-syn maintains good VFI performance on all the datasets with only slight PSNR/SSIM decrease compared to original SDL.

### 4.4 Applications beyond VFI

The proposed SDL achieves leading performance in VFI without using optical flows. It can also be used to address other CIT applications beyond VFI, such as image-to-image translation and image morphing. In this section, we take face aging and toonification and dog-to-dog image morphing as examples to demonstrate the generality of our SDL approach.

**Face Aging.** Unlike VFI, there is no public dataset available for training and assessing continuous I2I models. To solve this issue, we use StyleGAN [28, 29], which is a cutting-edge network for creating realistic images, to generate training data. Following [61], we use StyleGAN2 distillation to synthesize datasets for face manipulation tasks such as aging. We first locate the direction vector associated with the attribute in the latent space, then randomly sample the latent codes to generate source images. For each source image, we walk along the direction

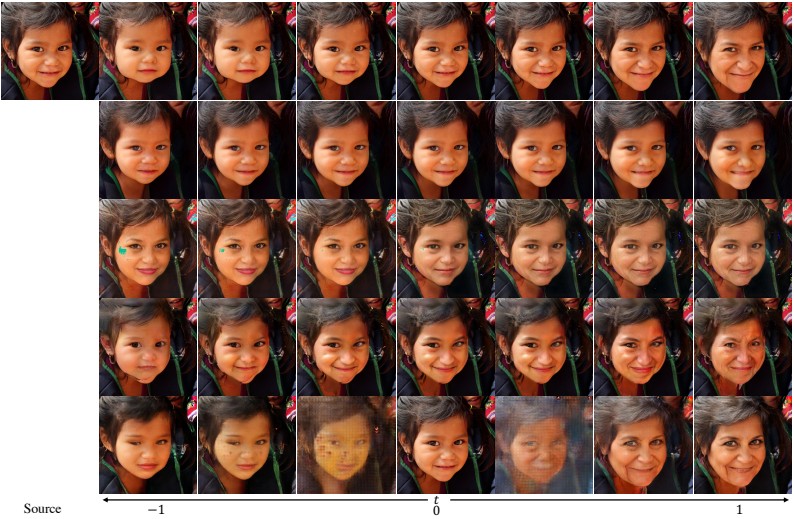

Source           $-1$           $t \atop 0$           $1$

**Fig. 6.** Comparison of SDL with competing methods on continuous face aging. From top to bottom: SDL, StyleGAN2 backpropagation [61], SAVI2I [36], Lifespan [43] and DNI [63].

vector with equal pace to synthesize a number of target images. As shown in the middle image of Fig.5, StyleGAN2 distillation may not always generate faithful images. We thus manually check all the samples to remove unsatisfactory ones. Finally, $50,000$ samples are generated, and each sample contains 11 images of $1024 \times 1024$. The dataset will be made publicly available.

The source image $I_0$ and a randomly selected target image $I_a$ ($a \in 1, 2, ..., 10$) are used as the inputs to train the SDL model. The corresponding value of $t$ is $a/10$. We also randomly replace the source image $I_0$ with the target image $I_{10}$ during training, and the corresponding value of $t$ can be set as $a/10 - 1$. In this way, the range of $t \in [0, 1]$ can be extended to $[-1, 1]$ so that our model can produce both younger (by setting $a \in [-1, 0)$) and older faces (by setting $a \in (0, 1]$). Note that SDL only needs the source image as input in inference.

Though trained on synthetic datasets, SDL can be readily used to handle real-world images. Since only a couple of works have been proposed for continuous I2I translation problem, and we choose those methods [63, 36, 43] whose training codes are publicly available to compare, and re-train their models using our datasets. In particular, we employ the same supervised $L_1$ loss as ours to re-train those unsupervised methods for fair comparison. Fig. 6 shows the results of competing methods on continuous face aging. One can see that SDL outperforms clearly the competitors in generating realistic images. By synthesizing the non-translatable features in reconstruction, SDL also works much better on retaining image background, for example, the mouth in the right-top corner. StyleGAN2 backpropagation [61] generates qualified aging faces; however, it fails to translate the face identity and loses the image background. SDL also produces more stable results than StyleGAN2 backpropagation, as shown in Fig.5.

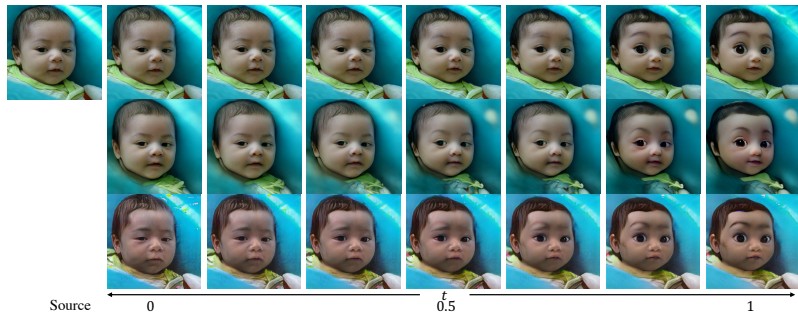

**Fig. 7.** Comparison of SDL with competing methods on continuous face toonification. From top to bottom: SDL, Pinkney *et al.* [46], and SAVI2I [36].

It is worth mentioning that SDL is $10^3$ times faster than StyleGAN2 back-propagation which requires time-consuming iterative optimization. SAVI2I [36] fails to generate qualified intermediaries with photo-realistic details. Lifespan [43] adopts an off-the-shelf face segmentation algorithm to keep the background unchanged. However, the generated face images have low quality. To test DNI [63], we train two Pix2PixHD [62] models to generate younger and older faces, respectively, and blend their weights continuously. As can be seen, DNI [63] fails to produce reasonable transition results. Moreover, SDL can generate continuous image-to-image translations with arbitrary resolutions, while all the competing methods cannot do it. More visual comparison results can be found in the **supplementary materials**.

**Face Toonification.** We first build a face toonification dataset by using the method of *layer swapping* [46]. Specifically, we finetune a pretrained StyleGAN on a cartoon face dataset to obtain a new GAN, then swap different scales of layers of the two GANs (*i.e.*, the pretrained and the finetuned ones) to create a series of blended GANs, which can generate various levels of face toonification effects. Similar to face aging, we generate $50,000$ training samples, each containing 6 images of resolution $1024 \times 1024$. During training, we take the source images (*i.e.*, $I_0$) as input and randomly choose a target image $I_a$, $a \in \{1, 2, ..., 5\}$, as the ground-truth output. The corresponding value of $t$ is $a/5$.

We compare SDL with Pinkney *et al.* [46] and SAVI2I [36], whose source codes are available. As shown in Fig. 7, SDL outperforms the competitors in producing visually more favourable results. Pinkney *et al.* [46] generates qualified toonification effects but it fails to retain the face identity and the image background. The generated face images of SAVI2I [36] have low quality. Furthermore, SAVI2I [36] merely synthesizes images with a resolution of $256 \times 256$, while SDL can yield results at any resolution. More visual comparison results can be found in the **supplementary materials**.

**Dog-to-Dog Morphing.** Similar to I2I translation, we synthesize training data for dog-to-dog morphing using StyleGAN2 [29] and BigGAN [7]. We randomly sample two latent codes as the source and target images. The intermediate images are obtained by interpolating the two codes in the latent space. We generate $50,000$ training samples, each containing 11 images of resolution $512 \times 512$.

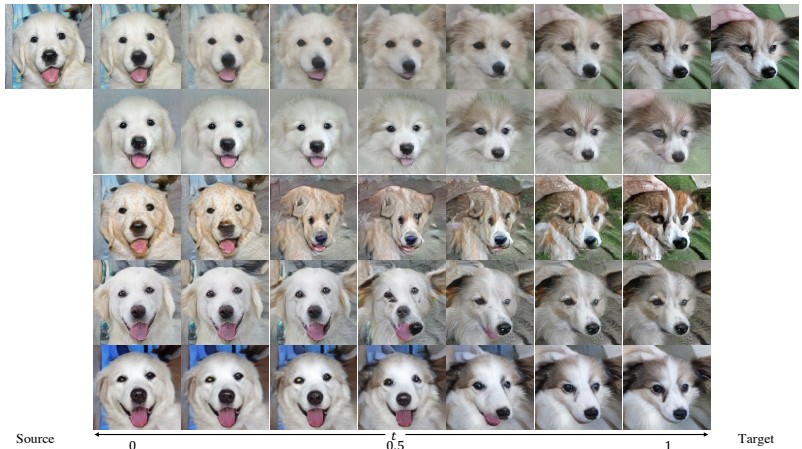

Source                                            $t$                                              Target
                        0                        0.5                        1

**Fig. 8.** Comparison of SDL with competing methods on dog-to-dog morphing. From top to bottom: SDL, StyleGAN2 backpropagation [61], CrossBreed [45], SAVI2I [36], and FUNIT [34].

During training, we take the source and target images (*i.e.*, $I_0, I_{10}$) as inputs and randomly choose an image $I_a$, $a \in \{1, 2, ..., 9\}$, as the ground-truth output.

Since few methods have been proposed for continuous image morphing, we compare SDL with I2I translation models, including CrossBreed [45], SAVI2I [36] and FUNIT [34]. (We re-train their models using our datasets and the same supervised $L_1$ loss for fair comparison.) As shown in Fig. 8, SDL achieves smooth morphing from one dog to another with vivid details. StyleGAN2 back-propagation [61] yields comparable results but it loses the background details. CrossBreed [45] and SAVI2I [36] fail to generate qualified intermediate results. FUNIT [34] produces smooth morphing; however, the generated dog images have low quality and it fails to retain the image content when $t = 0, 1$. Please refer to the **supplementary materials** for more visual comparisons.

## 5    Conclusion

We proposed a simple yet effective approach, namely space decoupled learning (SDL), for VFI problem. We implicitly decoupled the images into a translatable flow space and a non-translatable feature space, and performed image interpolation in the flow space and intermediate image synthesis in the feature space. The proposed SDL can serve as a general-purpose solution to a variety of continuous image transition (CIT) problems. As demonstrated by our extensive experiments, SDL showed highly competitive performance with the state-of-the-arts, which were however specifically designed for their given tasks. Particularly, in the application of video frame interpolation, SDL was the first flow-free algorithm that can synthesize consecutive interpolations with leading performance. In other CIT tasks such as face aging, face toonification and dog-to-dog morphing, SDL exhibited much better visual quality and efficiency with more foreground and background details.

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
