# OpenReview forum: "Beyond a Video Frame Interpolator: A Space Decoupled Learning Approach to Continuous Image Transition"
_thecvf.com/ECCV/2022/Workshop/VIPriors — VIPriors 2022 OralPosterTBD_

### Official Review · Reviewer_mcgQ · 2022-08-01
**The SDL method introduced in the paper is a solid contribution as it can improve Continuous Image Transition (CIT) and be deployed for several CIT tasks. The paper seems technically correct, the experiments are exhaustive, the motivation is clear, the paper is well written. Because of this, I vote to accept the paper.**

**Rating:** 8
**Confidence:** 4

**Review:**

Summary:

- The paper aims to improve video frame interpolation (VFI) by reformulate as a continuous image transition (CIT) task. The authors propose a model based on Space Decoupled Learning (SDL) that can be used for multiple CIT tasks, including VFI. One of the advantages of SDL is that is does not require human knowledge of the data domain. Thorough experiments show that the proposed SDL achieves competitive results on a number of CIT tasks, including VFI, face ageing, face tonification, dog-to-dog morphing.



Positive points:

+ The Space Decoupled Learning (SDL) for Continuous Image Transition (CIT) proposed in the paper seems a solid contribution and can be effectively deployed for several tasks, including VFI, face ageing, face tonification, dog-to-dog morphing.
+ The paper seem technically correct. Thorough experiments and ablation studies are performed to show the effectiveness of the proposed method. Experiments on four VFI datasets are included, on which SDL achieves competitive performance.
+ The paper is easy to read, the motivation is clear and the literature review exhaustive.
+ The proposed method is based on inductive priors which lead to decoupling the image space into tractable flow space and a non-tractable feature space, therefore this paper is a good fit for the VIPriors workshop.
+ The paper seems reproducible.



Negative points:
- I did not find substantial flows in the paper.



Per line comments:

225-234: In figure 1, the font size is very small, which makes the text hardly readable. I think the figure and capture are currently not self explanatory. I would suggest adding an explanation on how to read the figure.

315: In Table 1 the font size is very small. I suggest to increase it to make the table more readable.

---

### Official Review · Reviewer_LjJr · 2022-08-04
**.**

**Rating:** 8
**Confidence:** 4

**Review:**

1) Summary:

The paper reformulates Video Frame Interpolation (VFI) as a Continuous Image Transition task. The approach is based on Space Decoupled Learning (SDL) and it shows competitive results for VFI and for other CIT tasks.

2) Strengths:
- SDL simplifies VIF to a CIT problem without affecting performance and without requiring any human knowledge of the domain. Additionally, SDL works well for several CIT tasks.
- Paper very well written.
- Hypothesis supported by experiments.
- In line with workshop.

3) Weaknesses.
- Figure 1 is usually an important figure in the paper. It seems a bit difficult to follow and to read.

---

### Decision · Program_Chairs · 2022-08-08

**Decision:**

Accept (Oral/Poster TBD)

**Comment:**

Dear authors,


Congratulations! Your work has been accepted to the VIPriors workshop. Decisions on oral/poster presentations will follow later, when the program of the workshop is finalized.

*Please note the first action item is due on Wednesday! Please see instructions below.*

**Camera-ready instructions**

There is some work left to be done to ensure your work is included in the ECCV conference workshop proceedings. The ECCV publication managers use CMT to collect all workshop papers. This means we will migrate your paper from the VIPriors OpenReview page to the centralized ECCV workshop proceedings CMT page. The VIPriors program committee will ensure the details of your work (name, title, email address) are transferred to the CMT page, after which the ECCV proceeding managers will invite you to upload the camera-ready version of your work to the centralized ECCV CMT workshop proceedings page.

Please carefully follow the following instructions:
- **Before August 10th**, ensure that the first author has a CMT account under the same email address as the OpenReview account through which the accepted work was submitted. This account will be used to invite you to upload the camera-ready paper.
- Fill out this form, to inform us that the CMT account is in order: https://docs.google.com/forms/d/e/1FAIpQLSfyAoPv2_srESKaLRHIsHoWe3Fss1Z50ykdH7SzZpenA0m_5g/viewform
- Await instructions from the ECCV publication organizers, sent through CMT, on how to submit your camera-ready paper.
- Submit the camera-ready paper **before August 22nd**. Follow the camera-ready instructions for the main conference: https://eccv2022.ecva.net/submission/call-for-papers/.

**Attending the workshop**

We invite all authors of accepted works to attend the workshop in person on October 24th 2022 at ECCV in Tel Aviv. Please note a conference registration is required to attend the workshop. The workshop will be hybrid, enabling both in-person and remote attendance. We hope all accepted works can be represented in-person by at least one author, but we understand if this is not possible. Remote attendance of the workshop will be possible, though unfortunately there are limits on presenting works remotely: we intend to enable remote oral presentations, but this is not possible for posters.

Please fill out this form *before September 26th* to inform us of your attendance: https://docs.google.com/forms/d/e/1FAIpQLSfqRhdd2pq8t4CC8hL_c8fQo_TWcbzuQH3KGLzKVE36iTW_oQ/viewform.

**Presenting your work at the workshop**

Authors of all accepted papers are invited to present a poster at the workshop. Instructions on poster format will follow at a later date, but we will ask you to print and bring your own poster to the workshop.


For more information, as well as updates on the program of the workshop, keep an eye on our website: https://vipriors.github.io.

We thank you for choosing to submit to our workshop, and we are very much looking forward to hosting you in person in Tel Aviv!


Kind regards,

Robert-Jan Bruintjes
VIPriors program committee